# Diverse Co(II) Coordination Polymers with the Same Mixed Ligands: Evaluation of Chemical Stability and Structural Transformation

**DOI:** 10.3390/molecules29081748

**Published:** 2024-04-12

**Authors:** Chia-Yi Lee, Yu-Hui Ye, Song-Wei Wang, Jhy-Der Chen

**Affiliations:** Department of Chemistry, Chung Yuan Christian University, Chung Li, Taoyuan City 320, Taiwan; miss10031031@gmail.com (C.-Y.L.); a4862225@gmail.com (Y.-H.Y.); bvbv20520@gmail.com (S.-W.W.)

**Keywords:** coordination polymer, crystal structure analysis, structural transformation

## Abstract

Reactions of Co(OAc)_2_·4H_2_O, *N‚N’*-bis(3-pyridylmethyl)oxalamide (**L**) and 4,4′-sulfonyldibenzoic acid (H_2_SDA) afforded four coordination polymers with the same mixed ligands, {[Co(**L**)(SDA)(H_2_O)_2_]·H_2_O·CH_3_OH}_n_, **1**, {[Co(**L**)_0.5_(SDA)]·2H_2_O·0.5**L**}_n_, **2**, {[Co(**L**)_1.5_(SDA)(H_2_O)]·H_2_O}_n_, **3**, and {[Co_2_(**L**)_1.5_(SDA)_2_(H_2_O)_2_]·4H_2_O}_n_, **4**, which have been structurally characterized using single-crystal X-ray crystallography. Complexes **1**–**4** are 2D layers, revealing topologies of **sql**, 2,6L1, (4,4)Ia, and 6L12, respectively, and demonstrating that the metal-to-ligand ratio, solvent system, and reaction temperature are important in determining the structural diversity. The immersion of these complexes into various solvents shows that the structural types govern the chemical stabilities of **1**–**4**. Reversible structural transformation is shown for complexes **1** and **2** upon solvent removal and adsorption, while those of **3** and **4** are irreversible.

## 1. Introduction

The design and synthesis of functional coordination polymers (CPs) have drawn a great deal of attention from scientists over the past few years because of their intriguing structures and potential applications in the fields of sensors, gas adsorption and storage, heterogeneous catalysis, magnetism, and luminescence [1,2,3,4,5,6,7,8,9,10]. The structural diversity of CPs is conditional upon many factors, such as the identities of metal ions, linkers, and counter ions, as well as the reaction conditions involving the metal-to-ligand ratio, solvent system, and reaction condition. Therefore, efforts have been made to employ and control the applicable factors for preparing the target CPs.

Investigations into the structural transformations of CPs are important because such phenomena show potential applications in switches and sensors [11]. Structural transformations of CPs that lead to a change from one structure to another is difficult in the solid state due to the constrained rearrangement of the ligands. Structural transformation in CPs can be initiated by the removal of solvent, the exchange of guest molecules, exposure to reactive vapors, and external stimuli such as heat, light, and mechanical energy [12,13]. In spite of some advancement, great effort remains necessary to improve the capability to predict and control the structural transformation. Structural transformations in flexible bis-pyridyl-bis-amide (bpba)-based CPs supported by the polycarboxylate ligands have been reported [14,15,16]. To perform structural transformations upon solvent removal and adsorption for specific CPs with mixed ligands, it is required to prepare more than two CPs with the same metal ion, neutral spacer, and polycarboxylate ligands by manipulating the metal-to-ligand ratio, solvent system, and reaction condition.

In this work, we aim to elucidate the factors that govern the structural diversity of Co(II) CPs constructed from *N‚N’*-bis(3-pyridylmethyl)oxalamide (**L**) and 4,4′-sulfonyldibenzoic acid (H_2_SDA) (Figure 1) and to explore the roles of structural diversity in determining the structural transformations. The network structure of [Ag(**L**)_2_]NO_3_ [17] has been investigated and the pressure-induced polymerization of diiodobutadiyne with **L** has been reported [18]. Herein, we report the synthesis and structures of four diverse CPs, {[Co(**L**)(SDA)(H_2_O)_2_]·H_2_O·CH_3_OH}_n_, **1**, {[Co(**L**)_0.5_(SDA)]·2H_2_O·0.5**L**}_n_, **2**, {[Co(**L**)_1.5_(SDA)(H_2_O)]·H_2_O}_n_, **3**, and {[Co_2_(**L**)_1.5_(SDA)_2_(H_2_O)_2_]·4H_2_O}_n_, **4**, comprising mixed **L** and SDA^2−^, and an evaluation of their chemical stabilities and structural transformations forms the subject of this report.

## 2. Results and Discussion

### 2.1. Preparations for Complexes ***1**–**4***

While the procedures in the experimental section afforded the major products **1**–**4**, some minor products can also be obtained. Table 1 lists the reaction conditions as well as their yields, which are dependent on the metal-to-ligand ratio, solvent system reaction, and reaction condition.

### 2.2. Structure of ***1***

Crystals of complex **1** conform to the triclinic space group *P*ī, and each asymmetric unit consists of one Co(II) ion, one **L** ligand, one SDA^2−^ ligand, two coordinated water molecules, one co-crystallized water molecule, and one co-crystallized MeOH molecule. Figure 2a shows the coordination environment around the Co(II) metal center, which is six-coordinated by two nitrogen atoms from the two **L** ligands [Co-N = 2.1500(16)–2.1503(16) Å], two oxygen atoms from two SDA^2−^ ligands [Co-O = 2.0225(14)–2.0569(13) Å] and two oxygen atoms from the two coordination water molecules [Co-O = 2.1256(14)–2.1549(16) Å], resulting in a distorted octahedral geometry. The Co(II) central metal atoms are bridged by SDA^2−^ and **L** ligands to form a 2D layer. Topologically, if the Co(II) atoms are regarded as 4-connected nodes and the SDA^2−^ ligands as 2-connected nodes, while the **L** ligands are regarded as linkers, the structure of **1** can be simplified as a 2,4-connected 2D net with the (6^4^·8·10)(6)-2,4L3 topology (Figure 2b). Moreover, if the SDA^2−^ ligands are considered as linkers, the structure of **1** can be further simplified as a 4-connected 2D net with the (4^4^·6^2^)-**sql** topology (Figure 2c), as determined using the ToposPro program [19]. Complex **1** and {[Ni(**L**)(SDA)(H_2_O)_2_]·H_2_O·CH_3_OH}_n_ [20] are isostructural, indicating that the metal identity is not important in determining the structural topology of CP comprising **L** and SDA^2−^.

### 2.3. Structure of ***2***

Crystals of complex **2** conform to the triclinic space group *P*ī, and each asymmetric unit consists of one Co(II) ion, a half of an **L** ligand, one SDA^2−^ ligand, two co-crystallized water molecules, and half of a co-crystallized **L** ligand. Figure 3a shows the coordination environment around Co(II) metal centers, which are five-coordinated by two nitrogen atoms from the two **L** ligands [Co-N = 2.0600(13) Å] and four oxygen atoms from four SDA^2−^ ligands [Co-O = 2.0140(11)–2.0649(11) Å], resulting in distorted square pyramidal geometries. The Co(II) central metal atoms are bridged by SDA^2−^ ligands to form dinuclear Co(II) units [Co---Co = 2.7202(8) Å], which are further extended by **L** ligands to form a 2D layer. Topologically, if the Co(II) ions are regarded as 5-connected nodes and the SDA^2−^ as 4-connected nodes, while the **L** ligands are regarded as linkers, the structure of **2** can be simplified as a 4,5-connected net with the (4^6^·6^4^)(4^6^)-4,5L51 topology (standard representation), as illustrated in Figure 3b. If the dinuclear Co(II) units are regarded as 6-connected nodes and the SDA^2−^ ligands as 2-connected nodes, while the **L** ligands are regarded as linkers, the structure of **2** can be simplified as a 2,6-connected net with the (4^2^·6^8^·8·10^4^)(4)_2_-2,6L1 topology (cluster representation), as illustrated in Figure 3c. 

### 2.4. Structure of ***3***

Crystals of **3** conform to the triclinic space group *P*ī, and each asymmetric unit consists of one Co(II) ion, one and a half **L** ligands, one SDA^2−^ ligand, one coordinated water molecule, and one co-crystallized water molecule. Figure 4a shows the coordination environment around a Co(II) metal center, which is six-coordinated by three nitrogen atoms from the three **L** ligands [Co-N = 2.168(3)–2.228(3) Å], two oxygen atoms from two SDA^2−^ ligands [Co-O = 2.038(2) and 2.053(2) Å], and one oxygen atom from the coordinated water molecule [Co-O = 2.144(2) Å], resulting in a distorted octahedral geometry. The Co(II) central metal atoms are bridged by SDA^2−^ and **L** ligands to form a 2D layer. Topologically, if the Co(II) atoms are regarded as 5-connected nodes and the SDA^2−^ ligands as 2-connected nodes, while the **L** ligands are regarded as linkers, the structure of **3** can be simplified as a 2,5-connected net with the (4^2^·6^7^·10)(6) topology (Figure 4b). Moreover, if the SDA^2−^ ligands are considered as linkers, the structure of **3** can be further simplified as a 5-connected net with the (4^8^·6^2^)-(4,4)Ia topology, as illustrated in Figure 4c.

### 2.5. Structure of ***4***

Crystals of **4** conform to the triclinic system with the *P*ī space group. The asymmetric unit contains two Co(II) ions, one and a half **L** ligands, two SDA^2−^ ligands, two coordinated water molecules, and four co-crystallized water molecules. Figure 5a exhibits the coordination environment around Co(II) ions. While Co(1) is six-coordinated by one pyridyl nitrogen atom from the **L** ligand [Co-N = 2.133(3) Å], two oxygen atoms from two SDA^2−^ ligands [Co-O = 2.054(2)–2.118(3) Å], and one oxygen from the water molecule [Co-O = 2.147(2) Å], the Co(2) is six-coordinated by two pyridyl nitrogen atoms from the **L** ligand [Co-N = 2.167(3)–2.172(3) Å], two oxygen atoms from two SDA^2−^ ligands [Co-O = 2.063(3)–2.069(2) Å], and two oxygen atoms from two water molecules [Co-O = 2.118(3)–2.134(2) Å], resulting in distorted octahedral geometries. The Co(II) ions are bridged by the SDA^2−^ to form dinuclear units [Co---Co = 3.6113(7) Å], which are further extended by L to form a 2D layer. 

Topologically, if the Co(II) ions are considered as 5- and 6-connected nodes and the SDA^2−^ ligands as 2- and 4-connected nodes, while the **L** ligands are regarded as linkers, the structure of **4** can be simplified as a 2,4,5,6-connected 2D net with the (3^2^·4·5·6^5^·7)(3^2^·4^2^·5^2^·6^3^·7^2^·8^4^)(3^2^·6^2^·7^2^)(4) topology, showing cycle self-crossings (standard representation with 2-connected nodes), as shown in Figure 5b. If the 2-connected nodes of the SDA^2−^ ligands are considered as linkers, the structure of **4** can be simplified as a 4,5-connected 2D net with the (3^2^·4^2^·6^6^)_2_(3^2^·6^2^·7^2^) topology with crossing edges (standard representation without 2-connected nodes), as shown in Figure 5c. If the Co(II) dinuclear units are considered as 7-connected nodes and the SDA^2−^ ligands as 2-connected nodes, while the **L** ligands are regarded as linkers, the structure of **4** can be simplified as a 2,2,7-connected 2D net with the (4^3^·5^5^·6^8^·8^4^·10)(4)(6) topology (cluster representation with 2-connected nodes), as shown in Figure 5d. If the SDA^2−^ ligands are further considered as linkers, the structure of **4** can be simplified as a 6-connected 2D net with the (4^14^·6)-6L12 topology (cluster representation without 2-connected nodes) (Figure 5e). Complex **4** can thus be considered as a self-catenated CP.

Complexes **1**–**4** represent a unique example of four Co(II) CPs with **L** and SDA^2−^ ligands that have been structurally characterized. For a comparison, it is interesting to note that five Cd(II) CPs containing 1,4-bis(2-methyl-imidazol-1-yl)butane and 5-bromoisophthalate ligands have been reported [21].

### 2.6. Ligand Conformations and Coordination Modes

For the ligand *N,N’*-bis(3-Pyridylmethyl)oxalamide (**L**), the positions of the two C=O groups can be distinguished as *trans* or *cis*. When the two groups of C=O groups show opposite directions, they are defined as *trans*, and when they show the same direction, they are *cis*. Because of the different orientations adopted by the pyridyl nitrogen atoms and the amide oxygen atoms, three more conformations can be expressed as anti–anti, syn–anti, and syn–syn. Accordingly, the ligand conformations of **L** in **1**–**4** are listed in Table 2, along with the coordination modes of the SDA^2−^ ligands. Moreover, while the **L** ligands in **1**–**4** coordinate to the metal centers through the two pyridyl nitrogen atoms, the SDA^2−^ ligands bridge two and four metal atoms. 

### 2.7. Powder X-ray Analysis

In order to check the bulk purities of the products, powder X-ray diffraction (PXRD) experiments were carried out for complexes **1**–**4**. As shown in Appendix A, the peak positions of the experimental and simulated PXRD patterns are in agreement with each other, which demonstrate that the crystal structures are truly representative of the bulk materials.

### 2.8. Thermal Properties

Thermal gravimetric analysis (TGA) was carried out to examine the thermal decomposition of the four complexes. The samples were recorded from about 30 to 800 °C at 10 °C min^−1^ under a N_2_ atmosphere, Table 3 and Appendix A. Two-step decomposition was observed for the complexes, revealing that the decomposition temperatures for the frameworks of complexes **1**–**4** are in the range of 200–300 °C. 

### 2.9. Chemical Stabilities 

To check the chemical stabilities of complexes **1**–**4** in the various solvents, 10 mg of each complex was immersed in 10 mL of methanol (MeOH), ethanol (EtOH), ether, hexane, tetrahydrofuran (THF), acetonitrile (ACN), dichloromethane (DCM), dimethylacetamide (DMAC), and dimethylformamide (DMF), respectively, for a week, and then filtered and dried at room temperature. The PXRD patterns (Appendix A) show that complex **4** is unstable in all of the solvents, complex **1** is stable in EtOH, ACN, and DCM, complex **2** is stable in ether, hexane, and ACN, and complex **3** is stable in EtOH, ether, hexane, THF, ACN, and DCM, indicating that the structural diversity is important in determining the chemical stability of Co(II) CPs comprising **L** and SDA^2−^ ligands. 

### 2.10. Attempts for Structural Transformation

Complexes **1**–**4** provide an opportunity for an investigation of structural transformation due to the solvent exchange because they comprise the same metal centers and organic ligands with different metal-to-ligand ratios. To investigate the structural transformations, complexes **1**–**4** were first heated to 110, 130, 170, and 130 °C, respectively, for two hours to obtain fully desolvated samples. Appendix A depicts the colors of these complexes before and after heating. The PXRD patterns (Appendix A) demonstrate that the structures of these desolvated complexes were different from the original ones upon the removal of the crystallized solvents. Moreover, the desolvated product of **1** shows reversible structural transformations in MeOH, EtOH, and ACN, and that of **2** shows reversible structural transformations in the ether and DMAc solution. However, the desolvated samples of **3** and **4** in the solvents remained intact.

Complexes **1**, **3,** and **4** were all soluble in water, while complex **2** was slightly soluble in water. To investigate the stabilities of these complexes in water, complexes **1**–**4** were immersed in water for **2** days, and then the water was evaporated to obtain the solid samples. Their PXRD patterns and photos (Figure 6 and Figure 7) show that complexes **1**, **3,** and **4** and part of complex **2** were probably transformed into the same new complex. Their IR spectra confirm this change (Appendix A). Figure 7 depicts a drawing that summarizes the reaction pathways and colors of the complexes. 

## 3. Experimental Section

### 3.1. General Procedures

Elemental analyses were performed on a PE 2400 series II CHNS/O elemental analyzer (PerkinElmer instruments, Shelton, CT, USA). Infrared spectra were obtained from a JASCO FT/IR-460 plus spectrometer with pressed KBr pellets (JASCO, Easton, MD, USA). Powder X-ray diffraction patterns were carried out with a Bruker D8-Focus Bragg–Brentano X-ray powder diffractometer equipped with a CuKα (λα = 1.54178 Å) radiation (Bruker Corporation, Karlsruhe, Germany). Thermal gravimetric analyses (TGA) were carried out on a TG/DTA 6200 analyzer (SEIKO Instruments Inc., Chiba, Japan).

### 3.2. Materials

The reagent Co(OAc)_2_·4H_2_O was purchased from Alfa Aesar (Ward Hill, MA, USA), and the H_2_SDA from Aldrich Chemical Co. (St. Louis, MO, USA). The ligand N,N’-di(3-methylpyridyl)oxalamide (**L**) was prepared according to a published procedure [20].

### 3.3. Preparations

#### 3.3.1. {[Co(L)(SDA)(H_2_O)_2_]·H_2_O·CH_3_OH}_n_, **1**

A mixture of Co(OAc)_2_·4H_2_O (0.050 g, 0.20 mmol), **L** (0.027 g, 0.10 mmol), and H_2_SDA (0.031, 0.10 mmol) was placed in a 23 mL Teflon reaction flask containing 8 mL H_2_O and 2 mL MeOH, which was sealed and heated at 80 °C for 48 h under autogenous pressure. Then, the reaction system was cooled to room temperature at a rate of 2 °C per hour. Dark pink crystals suitable for single-crystal X-ray diffraction were obtained. Yield: 0.049 g (34%). Anal. Calcd for C_29_H_32_CoN_4_O_12_S (MW = 719.57): C, 48.40; H, 4.48; N, 7.79%. Found: C, 48.31; H, 4.11; N, 8.04%. FT-IR (cm^−1^): 3677 (m), 3504 (N-H, m), 3327 (w), 3060 (w), 1663 (C=O, s), 1599 (s), 1557 (m), 1512 (m), 1385 (s), 1295 (m), 1159 (s), 1102 (m), and 1036 (w) (Appendix A).

#### 3.3.2. {[Co(L)_0.5_(SDA)]·2H_2_O·0.5L}_n_, **2**

Complex **2** was prepared by following similar procedures as for **1**, except that a mixture of Co(OAc)_2_·4H_2_O (0.050 g, 0.20 mmol), **L** (0.054 g, 0.20 mmol), and H_2_SDA (0.031, 0.10 mmol) in 10 mL H_2_O was heated at 80 °C for 96 h. Indigo crystals were obtained. Yield: 0.035 g (26%). Anal. Calcd for C_28_H_26_CoN_4_O_10_S (MW = 669.52): C, 50.23; H, 3.91; N, 8.37%. Found: C, 50.50; H, 3.73; N, 8.29%. FT-IR (cm^−1^): 3504 (N-H, m), 3048 (w), 2944 (w), 1654 (C=O, s), 1597 (s), 1507 (s), 1404 (s), 1291 (m), 1227 (m), 1168 (m), 1103 (m), and 1034 (m) (Appendix A).

#### 3.3.3. {[Co(L)_1.5_(SDA)(H_2_O)]·H_2_O}_n_, **3**

Complex **3** was prepared by following similar procedures as for **1**, except that a mixture of Co(OAc)_2_·4H_2_O (0.050 g, 0.20 mmol), **L** (0.054 g, 0.20 mmol), and H_2_SDA (0.031, 0.10 mmol) in 10 mL H_2_O was heated at 60 ^o^C for 48 h. Dark pink crystals were obtained. Yield: 0.039 g (28%). Anal. Calcd for C_35_H_33_CoN_6_O_11_S (MW = 804.66): C, 52.24; H, 4.13; N, 10.44%. Found: C, 52.31; H, 3.94; N, 10.27%. FT-IR (cm^−1^): 3501 (N-H, w), 3394 (w), 3333 (w), 3275 (w), 1658 (C=O, s), 1599 (s), 1512 (s), 1393 (m), 1382 (m), 1102 (m), and 1034 (m) (Appendix A).

#### 3.3.4. {[Co_2_(L)_1.5_(SDA)_2_(H_2_O)_2_]·4H_2_O}, **4**

Complex **4** was prepared by following similar procedures as for **1**, except that a mixture of Co(OAc)_2_·4H_2_O (0.050 g, 0.20 mmol), **L** (0.027 g, 0.10 mmol), and H_2_SDA (0.031, 0.10 mmol) in 10 mL H_2_O was heated at 80 °C for 96 h. Violet crystals were obtained. Yield: 0.022 g (17%). Anal. Calcd for C_49_H_49_Co_2_N_6_O_21_S_2_ (MW = 1239.92): C, 47.46; H, 3.98; N, 6.78%. Found: C, 47.77; H, 3.69; N, 6.80%. FT-IR (cm^−1^): 3503 (N-H, w), 3255 (w), 2816 (w), 1662 (C=O, w), 1596 (s), 1514 (m), 1384 (m), 1351 (w), 1292 (m), 1228 (w), 1160 (m), 1100 (m), and 1034 (w) (Appendix A).

### 3.4. X-ray Crystallography

The single-crystal X-ray data of complexes **1**–**4** were collected using a Bruker AXS SMART APEX II CCD diffractometer equipped with graphite-monochromated MoKα radiation (0.71073 Å). They were then reduced using standard methods [22], followed by empirical absorption corrections based on a “multi-scan”. The direct or Patterson method was adopted to locate the positions of some of the heavier atoms, and the remaining atoms were established in a series of alternating difference Fourier maps and least-square refinements. Excluding the hydrogen atoms of the water molecules, those of the others were added by using the HADD command in SHELXTL 6.1012 [23]. Table 4 lists the crystal and structure refinement parameters for **1**–**4**. CCDC Nos. 2330673–2330676 contain the supplementary crystallographic data for this paper. These data can be obtained free of charge via http://www.ccdc.cam.ac.uk (accessed on 4 March 2024) or from the Cambridge Crystallographic Data Centre, 12 Union Road, Cambridge CB2 1EZ, UK; fax: +44-1223-336-033; e-mail: deposit@ccdc.cam.ac.uk.

## 4. Conclusions

By careful evaluation of the metal-to-ligand ratio, solvent system, and reaction temperature, four diverse CPs constructed from Co(II) salts, *N‚N’*-bis(3-pyridylmethyl)oxalamide (**L**), and 4,4′-sulfonyldibenzoic acid (H_2_SDA) were successfully obtained, displaying 2D layers with the **sql**, 2,6L1, (4,4)Ia, and 6L12 topologies, respectively. It has also been shown that the structural diversity is important in determining the chemical stabilities of complexes **1**–**4**. Reversible structural transformations were observed for complexes **1** and **2** upon solvent removal and adsorption, while those of **3** and **4** were irreversible. Moreover, complexes **1**–**4** decompose in water and may result in an identical product. 

## Figures and Tables

**Figure 1 molecules-29-01748-f001:**
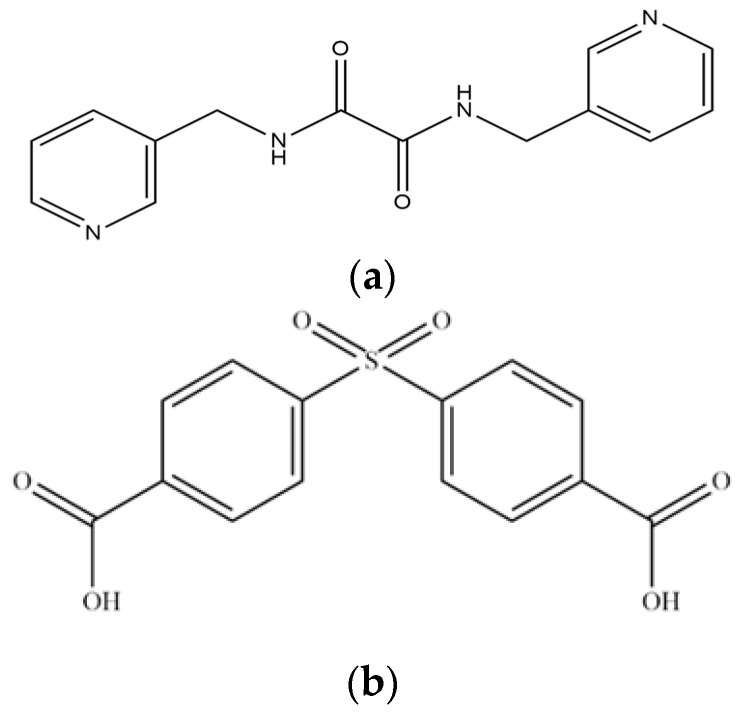
Structures of (**a**) **L** and (**b**) H_2_SDA.

**Figure 2 molecules-29-01748-f002:**
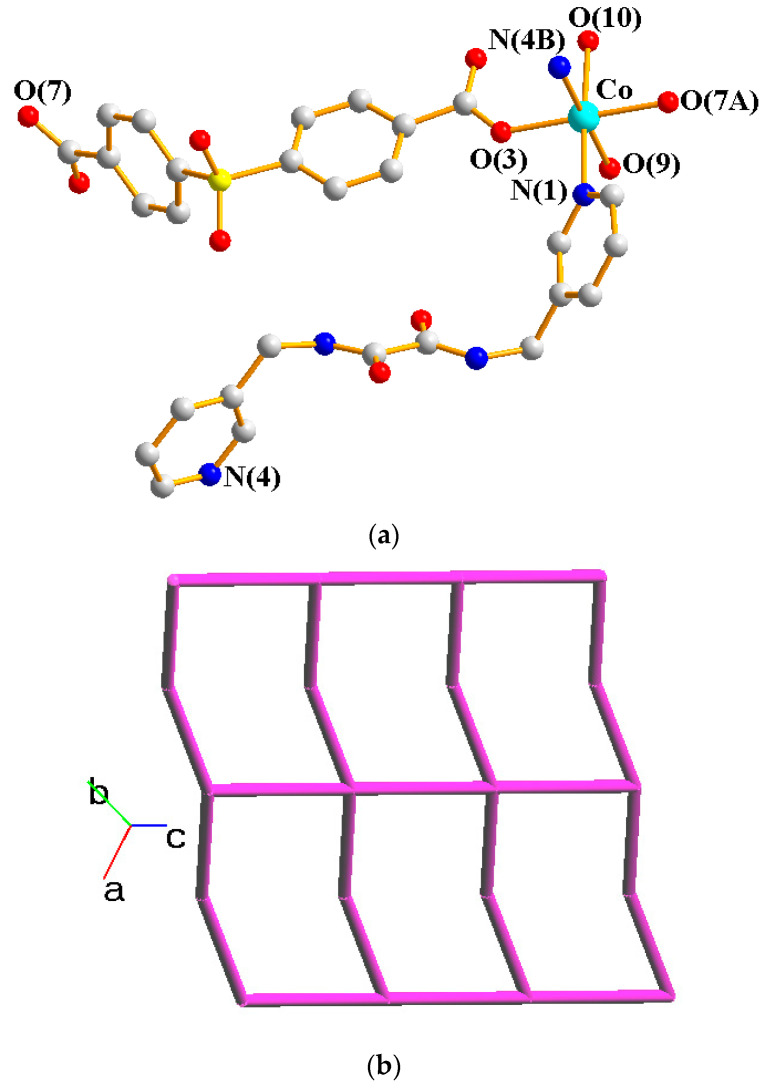
(**a**) Coordination environment of Co(II) ion in **1**. Symmetry transformations used to generate equivalent atoms: (A) x − 1, y, z − 1; (B) x, y, z − 1. (**b**) Topological structure of **1** with the (6^4^·8·10)(6)-2,4L3 topology. (**c**) Topological structure of **1** with the (4^4^·6^2^)-**sql** topology.

**Figure 3 molecules-29-01748-f003:**
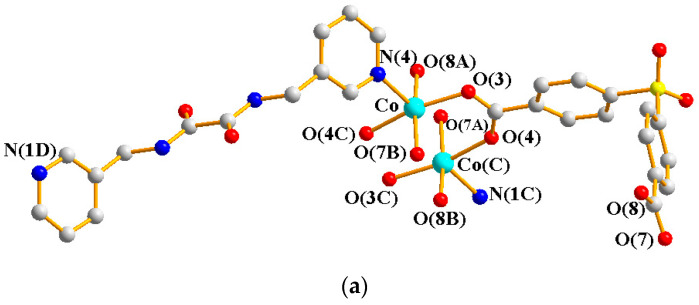
(**a**) Coordination environment of Co(II) ion in **2**. Symmetry transformations used to generate equivalent atoms: (A) x, y + 1, z − 1; (B) −x, -y, −z + 2; (C) −x, −y + 1, −z + 1; (D) −x + 1, −y + 2, −z + 1. (**b**) Topological structure of **2** showing the (4^6^·6^4^)(4^6^)- 4,5L51 topology. (**c**) Topological structure of **2** showing the (4^2^·6^8^·8·10^4^)(4)_2_-2,6L1 topology.

**Figure 4 molecules-29-01748-f004:**
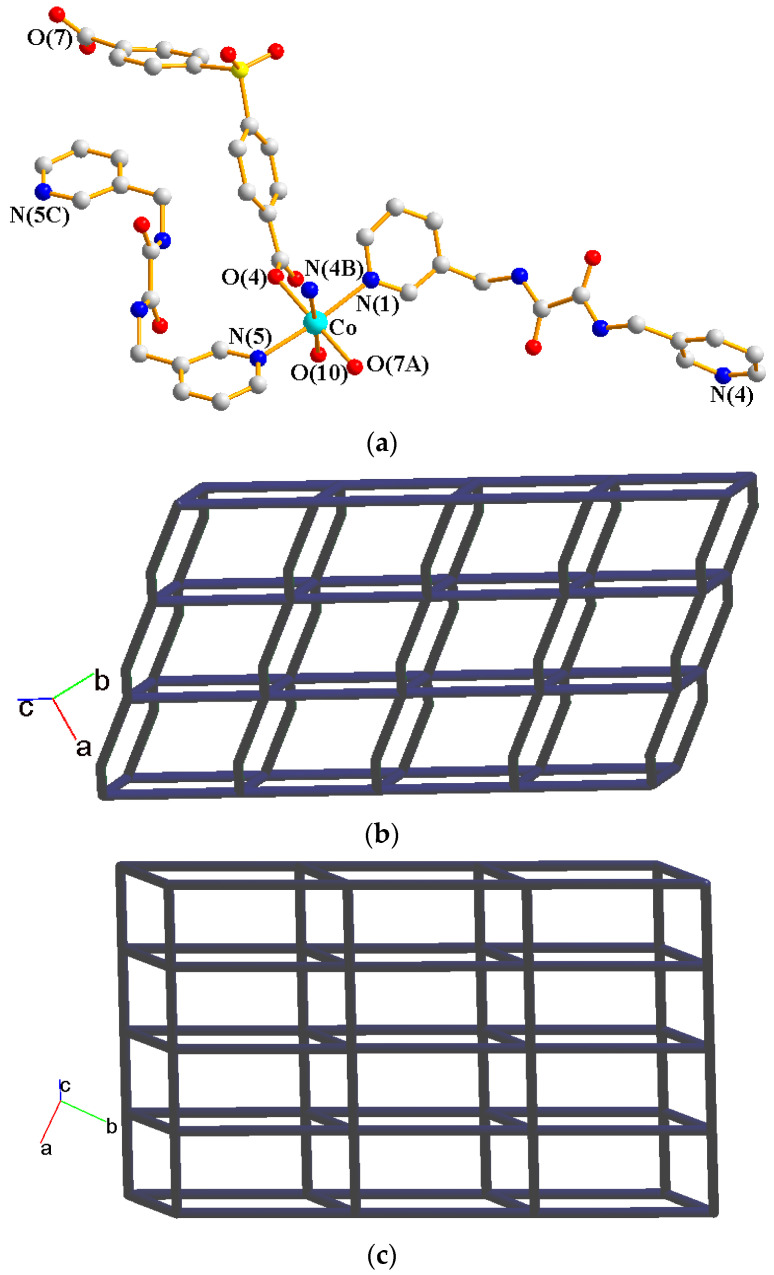
(**a**) Coordination environment of Co(II) ion in **3**. Symmetry transformations used to generate equivalent atoms: (A) x, y + 1, z + 1; (B) x, y, z − 1; (C) −x + 1, −y + 2, −z. (**b**) Topological structure of **3** with the (4^2^·6^7^·10)(6) topology. (**c**) Topological structure of **3** with the (4^8^·6^2^)-(4,4)Ia topology.

**Figure 5 molecules-29-01748-f005:**
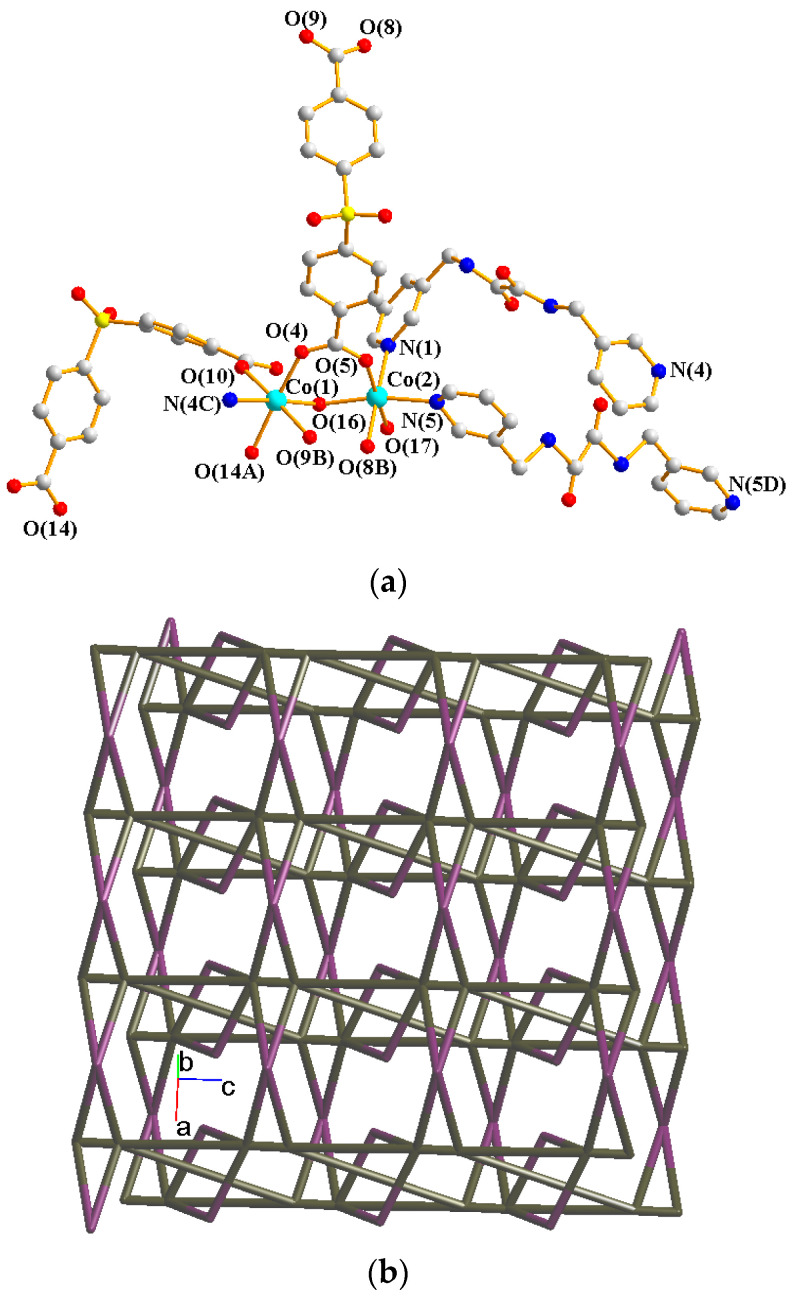
(**a**) Coordination environment of Co(II) ion in **4**. Symmetry transformations used to generate equivalent atoms: (A) −x + 2, −y + 1, −z − 1; (B) x + 1, y, z; (C) x, y, z − 1; (D) −x + 2, −y + 1, −z + 1. (**b**) Topological structure with the (3^2^·4·5·6^5^·7)(3^2^·4^2^·5^2^·6^3^·7^2^·8^4^)(3^2^·6^2^·7^2^)(4) topology. (**c**) Topological structure with the (3^2^·4^2^·6^6^)_2_(3^2^·6^2^·7^2^) topology. (**d**) Topological structure with the (4^3^·5^5^·6^8^·8^4^·10)(4)(6) topology. (**e**) Topological structure with the (4^14^·6)- 6L12 topology.

**Figure 6 molecules-29-01748-f006:**
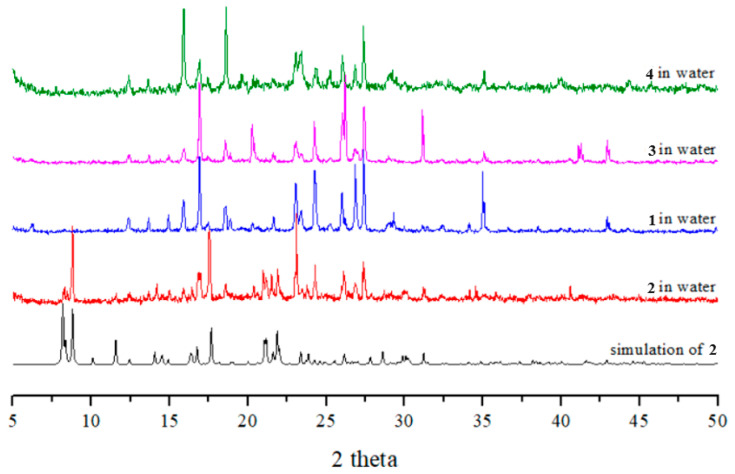
PXRD patterns and photos of complexes **1**–**4** after immersion in water.

**Figure 7 molecules-29-01748-f007:**
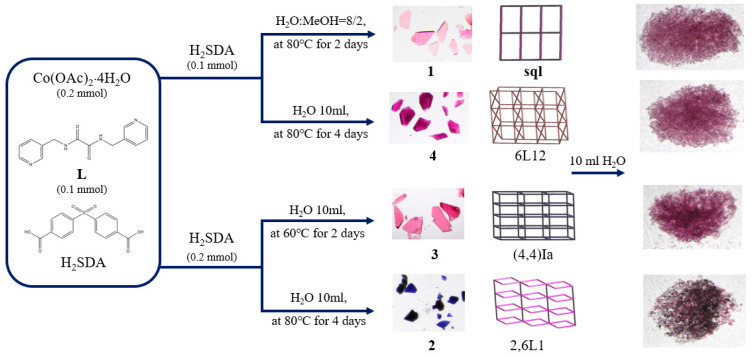
A drawing showing the reaction pathways and the colors of the complexes.

**Table 1 molecules-29-01748-t001:** Synthetic yields for complexes **1**–**4**.

0.20 mmol Co(OAc)_2_·4H_2_O + 0.10 mmol H_2_SDA + x L	Solvent	Temperature (°C)/Time (Days)	Complex	Yield (%)
x = 0.10 mmol	8 mL H_2_O and 2 mL MeOH	80/2	**1**	33.70
	10 mL H_2_O	80/2	**2** **3** **4**	3.29 ~0 13.55
		80/4	**2** **4**	2.09 17.34
x = 0.20 mmol		60/2	**3**	27.53
		80/2	**2** **3** **4**	4.18 9.28 6.45
		80/4	**2** **4**	26.21 6.86

**Table 2 molecules-29-01748-t002:** Ligand conformations and bonding modes of complexes **1**–**4**.

	Ligand Conformation	Coordination Mode
**1**	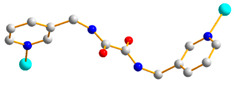 *trans syn-syn*	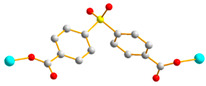 *μ_2_-κ*O:*κ*O’
**2**	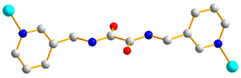 *trans syn-syn*	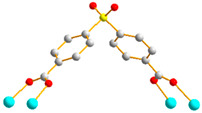 *μ_4_-κ*O:*κ*O’: *κ*O’’:*κ*O’’’
**3**	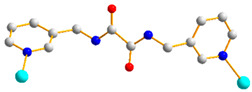 *trans anti-syn*	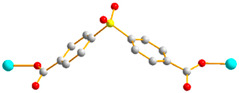 *μ_2_-κ*O:*κ*O’
**4**	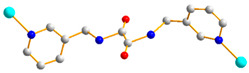 *trans syn-syn*	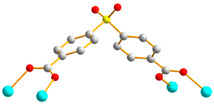 *μ_4_-κ*O:*κ*O’:*κ*O’’:*κ*O’’’
	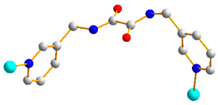	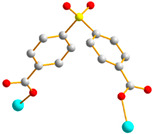
	*trans anti-syn*	*μ_2_-κ*O:*κ*O’

**Table 3 molecules-29-01748-t003:** Thermal properties of complexes **1**–**4**.

Complex	Weight Loss of Solvent °C (Calc/Found), %	Weight Loss of Ligand °C (Calc/Found), %
**1**	3 H_2_O + MeOH ~ 170 (11.95/9.23)	**L** + (SDA^2−^) 290–800 (80.06/82.72)
**2**	2 H_2_O ~ 130 (5.38/4.65)	**L** + (SDA^2−^) 270–800 (86.02/88.15)
**3**	2 H_2_O ~ 170 (4.48/4.56)	1.5 **L** + (SDA^2−^) 200–800 (88.36/85.91)
**4**	6 H_2_O ~ 130 (8.71/8.23)	1.5 **L** + 2 (SDA^2−^) 300–800 (82.02/78.76)

**Table 4 molecules-29-01748-t004:** Crystal data for complexes **1**–**4**.

Complex	1	2	3	4
Formula	C_29_H_32_CoN_4_O_12_S	C_28_H_26_CoN_4_O_10_S	C_35_H_33_CoN_6_O_11_S	C_49_H_49_Co_2_N_6_O_21_S_2_
Formula weight	719.57	669.52	804.66	1239.92
Crystal system	Triclinic	Triclinic	Triclinic	Triclinic
Space group	*P*ī	*P*ī	*P*ī	*P*ī
a, Å	12.0281(8)	10.537(3)	11.6913(8)	13.9967(9)
b, Å	12.3628(9)	12.378(2)	12.5564(8)	14.4949(8)
c, Å	12.5821(9)	12.849(2)	13.2862(8)	14.9094(8)
α, °	110.490(3)	61.476(8)	105.7159(18)	104.0893(14)
*β*, °	99.692(3)	72.697(10)	105.6037(18)	93.1620(15)
γ,°	108.754(3)	85.374(14)	91.3885(19)	114.1365(14)
V, Å^3^	1574.1(2)	1402.0(5)	1798.4(2)	2636.1(3)
Z	2	2	2	2
D_calc_, Mg/m^3^	1.518	1.586	1.486	1.562
F(000)	746	690	832	1278
µ(Mo K_α_), mm^−1^	0.682	0.754	0.605	0.795
Independent reflection	7756 [R(int) = 0.0276]	6938 [R(int) = 0.0202]	7082 [R(int) = 0.1016]	10,361 [R(int) = 0.0473]
Data/restraint/parameter	7756/0/452	6938/0/397	7082/0/497	10,361/2/776
Quality-of-fit indicator ^c^	1.048	1.063	1.004	1.019
Final R indices [I > 2σ(I)] ^a,b^	R1 = 0.0413, wR2 = 0.1070	R1 = 0.0296, wR2 = 0.0825	R1 = 0.0535, wR2 = 0.0871	R1 = 0.0495, wR2 = 0.1133
R indices (all data)	R1 = 0.0440, wR2 = 0.1089	R1 = 0.0323, wR2 = 0.0841	R1 = 0.1193, wR2 = 0.1061	R1 = 0.0811, wR2 = 0.1364

^a^ R_1_ = ∑∥F_o_∣ − ∣F_c_∥/∑∣F_o_∣. ^b^ wR_2_ = [∑w(F_o_^2^ − F_c_^2^)^2^/∑w(F_o_^2^)^2^]^1/2^. w = 1/[σ^2^(F_o_^2^) + (ap)^2^ + (bp)]. *p* = [max(F_o_^2^ or 0) + 2(F_c_^2^)]/3. a = 0.0477, b = 2.2049 for **1**; a = 0.0411, b = 1.2183 for **2**; a = 0.0299, b = 0.9541 for **3**; a = 0.04854, b = 5.8594 for **4**. ^c^ Quality of fit = [∑w(∣F_o_^2^∣ − ∣F_c_^2^∣)^2^/N_observed_ − N_parameters_)]^1/2^.

## Data Availability

The data are contained within the article or Appendix A.

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
