# Peer review of "Diverse Co(II) Coordination Polymers with the Same Mixed Ligands: Evaluation of Chemical Stability and Structural Transformation"

_molecules, 2024, doi:10.3390/molecules29081748_

Round 1

Reviewer 1 Report

Comments and Suggestions for Authors

In the submitted manuscript, the authors present research in the field of cobalt coordination polymers with the intention of demonstrating stability and structural transformations. Although this area is very interesting for research, the manuscript submitted is not suitable for publication. Given the significance of the entire study, experiments should be conducted at least in triplicate. Based on the manuscript, it is not clear whether the syntheses were performed only once or the robustness of the processes themselves was checked. The graphical presentations are not adequate, especially figures 2(a), 3(b), 3(c), etc. It is not necessary to draw representations of symmetric transformations. All this can be collectively presented as done in figure 7. The authors need to focus much more on understanding the transformations, explain in detail which phase is formed after the compounds are exposed to water. At the beginning of the paper, the authors mention the possibility of using the transformation in designing sensors and switches, but this possibility is not highlighted in the conclusion. I suggest that the authors continue to work on this issue and, after additional research on the transformations and clearer presentation of the results, resubmit the manuscript to the journal.

Comments on the Quality of English Language

No specific comment to English

Reviewer 2 Report

Comments and Suggestions for Authors

The article can be accepted for publication after the authors correct the following comments:

1)      It is necessary to add to the text of the article and conclusions a part with a discussion of the dependence of the resulting structure on crystallization conditions. In the text of the article this important issue is not discussed at all; the authors only provide a reference to the experimental part. In addition, please compare the structures with each other, supplementing with a discussion of the types of ligand coordination that is present in the text of the article. The stability of the complexes may also be related to differences in structures.

2)      In the conclusions, the authors mention a comparison with the obtained structures; it is necessary to provide a comparison in the main text of the article; in addition, the authors are silent about the obtained isostructural to compound 1 nickel(II) complex https://doi.org/10.3390/ijms23073603, which also necessarily to be mentioned.

1)      In the introduction, it is necessary to provide not only general information about CP, but also specific information about the ligands used in the work, compounds or features known from the literature.

Round 2

Reviewer 1 Report

Comments and Suggestions for Authors

Q1:  We thank the reviewer’s comment. If the synthesis for each complex was not performed more than three times, it was not possible to obtain the single crystal X-ray structure, elemental analysis and the other spectroscopic data, not to mention to investigate their chemical stability and structural transformation. These experiments were performed based on the single crystal obtained; the quantity of which is always low, although yield can be high. Thus, the synthesis should be repeated many times to complete the investigation and the single crystal obtained were rechecked by using the PXRD patterns to confirm they are of the same compounds. Synthetic yields for complexes 1 – 4 listed in Table 1 are given to show that the formations of the complexes are dependent on the metal to ligand ratio, solvent system reaction and reaction condition.

Question 1a: In light of everything the authors have written, how do they comment on the differences in yield for complexes 1-4 in Table 1? Particularly in the case of complex 3, when the yield is 0.0010%, this number doesn't make physical sense because such a small amount is impossible to determine. Please comment on this specifically.

In the experimental section, FTIR data are provided. It would be very beneficial if the authors, considering they have resolved crystal structures, could discuss the data and relate the vibrational bands to the structure.

Question 2: The authors need to focus much more on understanding the transformations, explain  in detail which phase is formed after the compounds are exposed to water.  

Ans: We thank the reviewer’s comment. The best way to explain in detail which phase is formed after the compounds are exposed to water is to obtain the single crystal structures of these complexes in water. However, in this investigation, it remains elusive to obtain the single crystals and thus the structural transformations were demonstrated by their PXRD patterns, IR spectra and colors, which are standard characterization for structural transformation. The different PXRD patterns indicate different phases of the CPs.

Question 3: Kindly request the authors to consider that an explanation of the transformation based solely on crystal structure represents a static model. It would be beneficial if the authors could comment on whether the transformation occurs in multiple steps, whether it involves a solid-state transformation, or if the mechanism includes a series of states leading through crystallization.

Question 4: For a comparison, it is interesting to note that 379 five Cd(II) CPs containing 1,4-bis(2-methyl-imidazol-1-yl)butane and 5-bro-380 moisophthalate ligands have been reported

Why do the authors mention cadmium coordination polymers, and how is this related to the current work?  If it's relevant, please provide more detailed references with additional data and conclusions.

Question 5:

Kindly request the authors to add details related to thermal experiments and microscopy.

Comments on the Quality of English Language

No comments

Reviewer 2 Report

Comments and Suggestions for Authors

The manuscript can be published after a minor revision:

Please provide a brief discussion of the dependence of structures on synthesis conditions, supplemented by a factual description of the resulting structures. The reader has to analyze the table with the experimental conditions himself without any description. Let me note that the reviewer does not ask to explain the thermodynamic stability of this or that structure and the reasons for its formation. Only facts, that were obtained. 
